# Innovation Capability and Open Innovation for Small and Medium Enterprises (SMEs) Performance: Response in Dealing with the COVID-19 Pandemic

Augustina Asih Rumanti [1,*], Afrin Fauzya Rizana [1], Lutfia Septiningrum [2], Rocky Reynaldo [3] and Mohammad Mi'radj Isnaini [4]

1   School of Industrial and System Engineering, Telkom University, Bandung 40257, Indonesia; afrinfauzya@telkomuniversity.ac.id
2   School of Information System, Telkom University, Bandung 40257, Indonesia; lutfiaseptiningrum@telkomuniversity.ac.id
3   Faculty of Business and Economics, University of Melbourne, Melbourne, VIC 3051, Australia; rreynaldo@student.unimelb.edu.au
4   Industrial Engineering Department, Bandung Institute of Technology, Bandung 40132, Indonesia; isnaini@itb.ac.id
*   Correspondence: augustinaar@telkomuniversity.ac.id

**Abstract:** The current business environment requires every organization or company to achieve optimal performance and maintain it. Innovation capability and open innovation practices play a critical role in improving organizational performance. However, their role in improving Small and Medium Enterprises (SMEs)'s performance, especially during the COVID-19 pandemic, still needs to be identified further. Thus, this study conducts empirical research elaborating intrinsic factors of innovation capability, as well as the influence of open innovation on organizational efforts, i.e., how SMEs achieve optimal performance during the COVID-19 pandemic. In this research model, 206 respondents were gathered and given a reearch questionnaire. The respondents are the owner of batik SMEs located in several regions in Indonesia. PLS-SEM is used to test the data, and the result of this study shows that all hypotheses developed in this study are accepted, i.e., SMEs' innovation capability and open innovation practices significantly influence financial and operational performance. The results show that in batik SMEs, the ability to innovate and open innovation, especially open entry innovation, can facilitate greater organizational performance. Therefore, batik SMEs woud benefit from initiatives and opportunities that improve their abilities in open innovation.

**Keywords:** innovation capability; open innovation; organizational performance

## 1. Introduction

In the current era of the COVID-19 pandemic that has hit the whole world globally, the implementation of open innovation for organizations is a challenge, especially because the pandemic has devastated not only economies, but also people's lives on a broader level. Such an impact has gone beyond what occured during the previous global economic crises [1]. Referring to the current situation, open innovation is seen as a critical aspect of economic recovery, both during and post pandemic [1].

Achieving an optimum level of performance is essential for businesses. Organizational performance can be identified through one's financial condition and ability to produce goods and services, i.e., operational performance [2]. In their strategy to innovate, businesses have engaged external parties to leverage internal capabilities [3]. Innovation can be defined as an organization's capability to leverage their entire resources to create a new capability and value [4]. The concept explains that by carrying out innovation activities, an organization can explore the use of every resource it has, both from internal and external to the organization. Innovations that are successfully carried out by an organization will

produce something new that adds value to the organization. [4,5]. Developing innovation capability becomes important, for innovation plays a key role in determining the longevity and growth of the organization [3,6]. The capability to develop ideas is one of the priorities in an organization. Ideas developed are all ideas that exist in the organization, both ideas from employees for routine operational activities, as well as from executives related to short and medium-term strategic activities in the organization. While the ideas of top managers are related to the development and long-term strategy for the organization [7]. There are several innovation concepts for organizations, namely top-down, bottom-up, and outside-in [8]. Each of these concepts can be implemented according to the conditions or needs of the organization [9]. The concept of innovation in an organization is also influenced by the culture of the organization. Top-down innovation is an innovation that originates or is proposed by the leadership of the organization, which is then formally communicated to every element in the organization [8,9]. Top-down innovation requires a very high level of management involvement, which will drive future organizational performance, while bottom-up innovation allows existing and managed innovations to come from every party in the organization and allows every member to participate in the innovation. [8,9]. Outside-in innovation is an innovation concept that adopts or brings innovation sources from outside to enter the organization through social networking activities [8]. In this study, the innovation concept developed is a collaboration of the three concepts there are top-down, bottom-up, and onside-in. That collaboration will support the implementation of open innovation in achieving more optimal organizational performance, especially during the current pandemic. The surge of the knowledge economy has intensified global competition, and simultenously technological advancement has pushed innovation into the spotlight of business competition [7,10].

Innovation is a mechanism by which an organization produces new goods, processes, and systems necessary to adapt to changes in the market, technology, and competition mode [6,7,10]. Innovation is a complex activity that involves the production, diffusion, and translation of knowledge in the form of new or modified products or services; or the development of new production or processing techniques [11].

Successful innovation could bring uniqueness unavailable to other organizations, which becomes a source of advantage [12,13]. Innovation can also be seen as a process to increase the organization's must-have capabilities regardless of the organization's scale [12]. Innovation pertains to a continuous improvement within the organization to optimize its performance [14]. The ability to innovate is a critical success factor for the company's future growth and performance, and is one of the main ways companies can thrive in the midst of competition and gain profits. Strategy to innovation activities, along with a shared innovation vision, is critical when creating innovation capabilities. Innovation capability is a theoretical framework that can describe actions to support and increase the success of innovation activities [15].

Businesses, including SMEs, have been facing challenges in creating a balance between economic, social, and environmental factors [16]. SMEs are one of the business entities that make a significant contribution to economic conditions in Indonesia. This is the reason why information about the condition and presence of SMEs is important data for the government as an indicator for Indonesia's economic conditions, especially during the COVID-19 pandemic. The Indonesian Statistic Center is the government agency that manages the data. Data obtained from the Indonesian Statistic Center in 2020 is one of the bases for obtaining information on the condition of SMEs during the COVID-19 pandemic. Through this data, SMEs can be identified early that can survive or are forced to close or stop their activities [17]. According to the Statistic Center Indonesia 2020, there was a decrease in the number of SMEs in data collection locations by 7.06%, and by the end of 2020, there were 11.25% of SMEs that stopped operating [17]. SMEs need a strategy to innovate optimally [18]. In open innovation, knowledge used to facilitate innovation can be obtained externally or internally [18–20]. The stakeholders involved are suppliers, consumers, competitors, and the public. Open innovation explores sources of innovation from the outside (inbound) and

the inside (outbound), which then be used to accelerate internal innovation, expand the market, and create innovation for external parties [19]. SMEs are involved in many open innovation activities, especially for matters related to the market, such as meeting customer demands or competing with competitors. This has become more and more ubiqutious, especially in the face of the fourth industrial revolution [20]. Open innovation activities are carried out by exploring sources of innovation, both from outside (inbound) and from inside (outbound), which are then used to accelerate internal innovation, expand markets, and create innovations for external parties [21]. These activities are carried out by SMEs in many places, especially to support industry 4.0 [22].

This study was conducted to accommodate the current global pandemic because, during this time, SME owners or leaders need to find ways to maintain their business in a sustained manner, especially for SMEs in developing countries [21]. SMEs are considered an important factor in driving the economy and are one of the indicators of economic development of a country [22,23]. SMEs make a significant contribution to a country's GDP, both for developed and developing countries [21–23]. Therefore, the performance of SMEs is an important thing to study during a pandemic to determine necessary measure that can support them. In Indonesia, SMEs are one type of businesses that is quite vulnerable in facing the economic consequence of the current pandemic. On the other hand, SMEs are considered labor intensive, so the existence of SMEs can reduce unemployment, especially because of the economic impact of the economic crisis due to the pandemic [21]. Seeing the importance SMEs both in developing and developed countries, the performance of SMEs needs to receive a more in-depth attention and study, especially in the current pandemic condition. SMEs are the backbone of a country's economy [24], and their performance is one of the factors that measure the success of economic activities in a country [22,24].

Open innovation only occurs when organizations, in this case SMEs, collaborate actively with other parties and contribute to the market exploitation, market test, and demand analysis [2]. Open innovation currently has a paradigm that adopts several influences from the existence of digital technology and the global transformation of information and knowledge. This open innovation was later known as open innovation 4.0 [25]. Industry 4.0 is relevant for SMEs because with open innovation 4.0 in SMEs, access to sources of innovation will become wider which allows every stakeholder to be directly involved in the development of SMEs. Open innovation 4.0 provides space for all stakeholders to contribute more optimally to the company or organization. The transformation of information and knowledge, as well as its digitization, is important to the implementation of open innovation 4.0. Therefore, global collaboration networks and activities significantly impact open innovation 4.0. The open innovation 4.0 framework considers customer community as an important factor in the improvements made by a company or organization through open innovation activities. The involvement of the consumer community will accelerate the process of improvement and open innovation because it will increase the awareness of a company or organization with regards to the importance of innovation process. The company will develop a new capability to engage the consumer community through digital media to collect and analyze data that is part of a dynamic ecosystem in a value chain [25]. This affects the open innovation process for both large, medium, and small-scale companies, including SMEs.

In general, SMEs have difficulty identifying business possibilities outside of their core competencies. This can happen because of a focused product portfolio, a specific knowledge base, and limited financial resources. SMEs can strengthen their position by implementing open innovation [26]. SMEs are more dependent on external knowledge than large companies. Therefore, SMEs would benefit from collaboraiton with external parties to carry out an open innovation process [27]. These collaborative activities can increase the value created from innovative activities [28]. With limited resources, SMEs need to find a way to achieve economic industry in production, effectively market their products and offer support services that meet customers' needs. Collaboration with external parties is one of the ways that SMEs enable open innovation [2,3,27]. Open innovation

can be understood as the antithesis of the traditional vertical integration model, where Internal innovation activities lead to outputs in the form of products and services developed internally; these products and services are then distributed by the company to external parties [29]. Individual characters in an organization or company will have an impact on the innovation process [30], including the process and implementation of open innovation. Individual attitudes and motivations in innovating will affect the results of open innovation carried out by the company [31]. The process of adopting innovation by individuals or employees in an organization or company contributes to the organizational culture. Open innovation based on organizational culture will help organizations to seize new opportunities. This is because human or individual factors have a role in determining the success of the innovation process, hence the human factor cannot be ignored in order so that optimal innovation process can take place [31].

The challenge for SMEs to optimize their performance has become even more difficult with the rise of the global pandemic of COVID-19 [32]. This happens to all organizations across the economic scale. In the innovation process of an organization or company, the adaptability of individuals or employees in the company is important, especially during the implementation of new technologies and in sustaining the ability to compete in today's dynamic market. During the COVID-19 pandemic, the implementation of open innovation is a challenge for organizations or companies. The negative impact of the pandemic is not only on the economic system but also on various social aspects of society that may not have been imagined before will become a global crisis [1]. For SMEs, the role of individuals is very important in determining the readiness and success of these SMEs to carry out open innovation, especially the ability to respond efficiently to international market dynamism. Thus, Human capital is essential for the development and commercialization of innovations, and plays a key role in the survival and success of the innovation process, especially in the face of today's seemingly dynamic market environment [33].

SMEs need to work their way out in accessing resources necessary for implementing their innovation activities, particularly in open innovation [34]. Prior research argued and described the influence of innovation capability on organizational performance [3,6]. Yet there seems to be a lack of elaboration regarding factors that support innovation capability and the innovation needed to enhance structure performance within the context of the pandemic. Limited access and activities during the pandemic need to be mitigated by virtually optimizing the supporting factors of open innovation, e.g., implementing information technology and information management system [35,36].

Several previous research explained that innovation capability has an influence on the performance of an organization [3,6], and empirically open innovation has an influence on organizational performance, both in the form of operational performance or financial performance [2,27]. Therefore, this study was conducted to show the independent influence of each construct, namely innovation capability and open innovation, on the performance of the organization/company. Innovation capability is an important factor in organizational or company performance, especially financial performance [37], and significantly influences innovation performance in organizations or companies [38]. There is a study that explains the weak relationship between the construct of innovation capability to open innovation, which is less than 0.5 [39], and the relationship between the construct of open innovation to performance with a correlation of 0.16 [39]. At the same time, other studies explain a strong relationship or influence between innovation ability and organizational performance [40]. Therefore, in this study, a research model was constructed in which innovation capability and open innovation were directly related to organizational performance. Two constructs are independent variables on the basis that each construct has an influence on the performance of the organization or company, as has been conveyed through the model elaboration scheme in Figure 1.

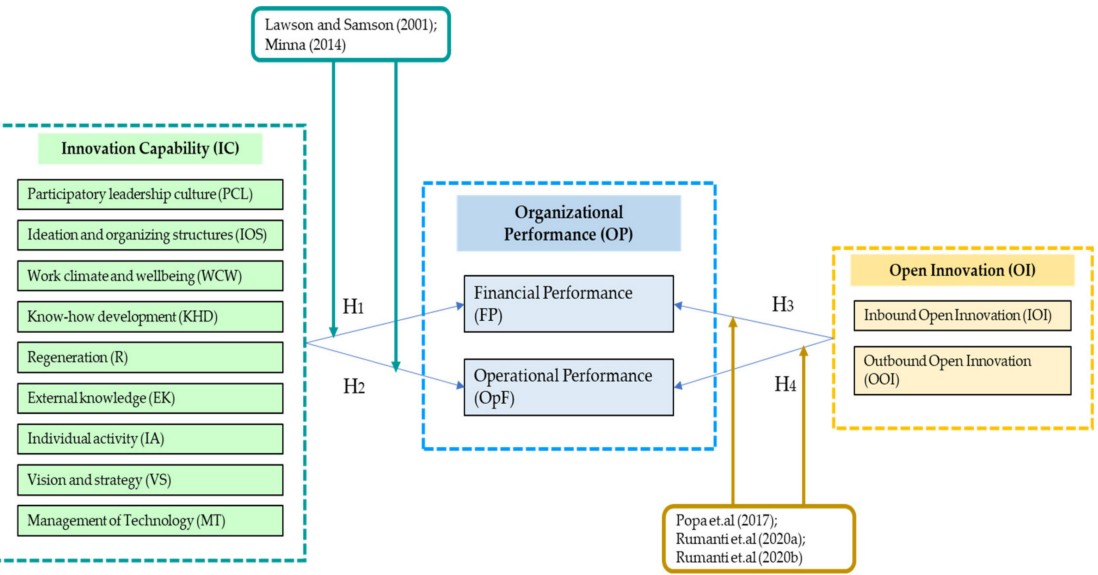

**Figure 1.** Elaboration scheme for research model.

Considering such a gap, this research conducts an empirical study elaborating intrinsic factors on innovation capability as well as the influence of open innovation on SME's efforts to attain optimal performance. The next section will lay out the literature study and hypotheses followed by the research methodology with a total of 206 respondents from batik SMEs. An analysis of the data is then conducted, and finally, a discussion on research findings, research limitations, and the overall conclusion.

## 2. Literature Review

Effective and measurable performance requires a series of preparation by the management team. Open innovation that is well prepared would facilitate and affect organizations in improving their performance [41]. This literature review will be divided into three parts according to the main constructs, i.e., Firm Performance, Innovation Capability, and Open Innovation.

### 2.1. Firm Performance

There are several concepts that are used in measuring firm performance, i.e., operational performance and financial performance [2,3]. Operational performance is usually measured as a set of several dimensions that reflect the internal operations of an organization in terms of the elements of the product, process quality, efficiency, and productivity. In some studies, operational performance was measured through the productivity, effectiveness, and efficiency of internal operations, while financial performance is measured through measures that include profitability, return on investment, and share price [3,6]. Increasing innovation, especially in financial performance, has an impact on the company's competitiveness. The competitiveness of a company can be supported by effective performance measurement. As a consequence, the company must know the factors that affect its performance and manage it effectively [11,42,43].

### 2.2. Innovation Capability

In an organization or company, innovation capability integrates the organization's key capabilities and resources to stimulate innovation successfully in an effort to achieve optimal organizational performance [44]. Innovation capability also reflects the ability to continuously transform knowledge and idea into a new product, process, and system for the benefit of the organization as well as its stakeholders. Innovation capability can be seen as a dynamic capability that shows an organization's ability to integrate, build, and reconfigure both internal and external competence to deal with a constantly changing environment [45].

Innovation capability also acts as a funnel that can seek, locate and develop potential innovations to be streamed to the core process in the organization [6]. A high degree of innovation capability enables the organization to bring efficiency together with creativity [6] and helps businesses to have the ability to create and develop excellent new products; this capability is a critical in order for businesses to survive and prosper in a long term [46]. That is why innovation capability is regarded as a valuable asset for firms in terms of providing and sustaining competitive advantage and implementing corporate strategy; it also helps the organization to form, manage, and integrate multiple capabilities or stimuli to innovate successfully [43]. Innovation capability is considered one of the factors that influence the policies of an organization, including SMEs [27]. Several studies have proposed direct relationship of innovation capability to organizational performance [3,38,47,48]. It indicates that innovation capabilities determine firm performance [49]. Thus, the development and analysis of indicators in innovation capabilities is important in facilitating performance evaluation in SMEs [50,51]. It is critical because innovation is believed to be the most important driver of organizational performance and serves as a key role in the survival and growth of an organization [3]. It is believed that innovative firms would have better firm performance compared to those that do not innovate. Innovative organizations show a higher level of economic growth and productivity than non-innovative ones [37]. The organizational performance itself can be measured through two dimensions, namely financial performance, and operational performance [2]. Moreover, the study provided a positive relationship between organizational innovation capability and performance [3,6]. Based on the explanation of the results of previous studies, the following hypotheses can be formulated:

**Hypothesis 1.** *There is a relationship between innovation capability and a firm's financial performance.*

**Hypothesis 2.** *There is a relationship between innovation capability and a firm's operational performance.*

*2.3. Open Innovation*

Open innovation refers to the inflow and outflow of knowledge activities undertaken by an organization to accelerate internal innovation, expand the market as well as use external innovation [27]. Open innovation illustrates how valuable ideas can be obtained from the internal and external parts of organizations, including from the market. The approach of such a paradigm plays a role in understanding the market demands [18,52,53]. Open innovation could be considered a contemporary paradigm for implementing innovation. Within open innovation, there are several forming components such as shared creativity, collaboration, and the attainment and management of the latest knowledge [54]. Open innovation is a way for organizations to increase their innovation capability in terms of technology and competition [55]. An organization can gain knowledge from the external environment to improve its capabilities in doing business and in managing existing knowledge [54,55]. Open innovation is one of the effective driving forces to encourage improving the performance of an organization. The development of open innovation can maintain the company's level of innovation and continuous innovation momentum so that an organization or company can face the increasingly fierce industry competition [56].

The occurrence of an open innovation process within SMEs could change the organizational system for the better, i.e., improvement in collaboration, knowledge among employees, and market knowledge [57]. Open innovation entails an internal and external openness that is relevant to creating innovation. Some of the benefits of the knowledge flow to and from the organization include encouraging internal innovation, expanding markets, and creating external innovations for other parties [27].

Open innovation is divided into two parts: inbound open innovation and outbound open innovation [2,58]. Inbound open innovation is realized by exploring the sources of

innovation such as information and new technology from external parties, e.g., customer, supplier, competitor, government, consultant, university, or research organization [25]. Outbound open innovation is realized by developing internal innovation capabilities so that the result can give to external organizations through licensing, patents, or contracts to gain financial and non-financial benefits [19,27]. Open innovation is one of the requirements for an organization to achieve optimal performance, especially for financial and operational performance [2]. Based on the compiled literature on open innovation, the third and fourth hypotheses were constructed:

**Hypothesis 3.** *There is a relationship between open innovation and a firm's financial performance.*

**Hypothesis 4.** *There is a relationship between open innovation and a firm's operational performance.*

The elaboration scheme for the research model that consists of these three constructs is shown in Figure 1.

### 3. Research Methodology

This section will elaborate on the framework by which the research was conducted. There are four sections, i.e., sample and data collection, the process of developing the research instrument, the description of research variables, and the process of data analysis. These stages are carried out to be able to explain in detail the flow of the methodology in this research.

#### 3.1. Sample and Data Collection

There are two stages in the data collection process. The first stage is a preliminary study (pilot test) to test the validity and reliability of the measuring instrument in this study. In this preliminary stage, questionnaires were distributed to 30 respondents/SMEs who were randomly selected from all data on SMEs batik Lasem, Rembang Region, Central Java Province, and Madura, East Java Province. Based on the responses to the questionnaire obtained in the preliminary study, several improvements were made to the questionnaire. Based on the initiation of the questionnaire that has been collected, several statements or questions must be modified in terms of the phrase structure in the narrative explanation of each statement. This improvement is made so that respondents can unequivocally understand the intent of the questions asked.

The population in this study were all SMEs in Lasem, Rembang Region, Central Java Province, and Madura, East Java Province. The characteristics of the respondents are SMEs affected by the COVID-19 pandemic, i.e., SMEs whose sales results, number of employees, and asset values have continued to decrease. In this study, the data collection technique used is purposive sampling, where the aim is to obtain a sample that can describe the population and save costs and time. The number of samples taken from the population is five to ten times the number of variables used in the analysis design and a minimum of 200 samples [57]. The number of questionnaires collected is that the general response was stuffed out by the owner of the batik SMEs who was respondents during this study. The distribution of questionnaires was carried out during June-August 2021. A 224 completes questionnaires were received out of a total of 242 questionnaires distributed. The response rate for this study was 92.56%. Meanwhile, based on the results of checking the standard of filling out the questionnaire, of the 224 questionnaires received, only 206 (85.12%) of the questionnaires met the need for further processing. The response rate of 92.56% can be achieved by distributing questionnaires directly to respondents, both respondents in Madura and in Rembang. Respondents were given guidance and assistance when filling out the questionnaire, so that most of the questionnaires could be filled out completely. However, respondents still have full authority and awareness to be able to fill out the questionnaire according to the actual conditions without the influence of the researcher. Unused answers are answers from respondents who do not meet the factors used in the

data processing. Answers that do not match the criteria can occur because the respondent does not fill out the questionnaire completely, either it does not match the number of questions or because it adds answers to the questionnaire given. In this study, there were 12 (4.96%) questionnaire responses that were not used due to the questionnaire responses not being filled out completely.

### 3.2. Instrument Development

In this study, the data used were based on the results of the distribution and filling of qualitative questionnaires. The questionnaire used in this study is presented detail in Appendix A. To ensure that the data collected is in the form of quantitative data, the answers to the questions in the questionnaire use a Likert scale from number 1 to number 5 [59]. The Likert scale 1 indicates strongly disagree, whereas the Likert scale 5 indicates strongly agree. The selection of the Likert scale was carried out with the thought that the Likert scale was more appropriate for the overall type of respondent, particularly by using a scale of 5 [59,60].

### 3.3. Operational of Construct

Innovation capability and open innovation as independent variables and organizational performance as the dependent variable.

### 3.3.1. Independent Variable

The independent variables in this study consist of innovation capability and open innovation. The innovation capability variable has nine dimensions that reflect the innovation capability. These dimensions are participatory leadership culture, ideation, and organizing structures, work climate and wellbeing, know-how development, regeneration, external knowledge, individual activity, vision and strategy, and management of technology. Each open innovation variable has two dimensions, namely inbound open innovation and outbound open innovation. Both dimensions reflect the open innovation variable.

### 3.3.2. Dependent Variable

In this study, the dependent variable is organizational performance. Organizational performance can be reflected through two dimensions in it, namely financial performance and operational performance. Every organization needs to be able to harmonize financial performance and operational performance because these two dimensions influence the achievement of organizational performance [2].

### 3.4. Data Analysis

In this study, PLS-SEM was used to test the data. In this research model. Using the 10:1 ratio as advised by Hair et al., the minimum variety of samples required has been met during this study. The small sample size does not influence PLS-SEM as a result of this system only analyzing one construct at a time by applying an iterative order of ordinary squares and multiple linear regression [61].

In estimating the importance of the relationship, PLS-SEM uses a boot-strapping approach that does not need constant assumptions. Therefore, PLS-SEM is appropriate for analyzing sample data with small amounts and is not normally distributed [62]. In addition, PLS-SEM also does not require homogeneity in the processed data [63]. The data used in this study were 206 respondents from two observation locations with different distributions, namely 106 respondents in Madura and 100 respondents in Lasem.

The character of each object of research can be said to be similar because of the similarities in carrying out the observed process, namely the process of making batik. The measurement model is often evaluated based on four criteria, namely composite reliability to measure internal consistency reliability, outer loading, and average variance extracted (AVE) to judge convergent validity, as well as cross-loading and Fornell-Larcker values to evaluate discriminant validity [62]. Cronbach's alpha indicator, which is based on

indicator intercorrelation with the assumption that all indicators have the same outer loading on the construct, can check the consistency reliability criteria. Another criterion used is composite reliability which considers different outer loadings for every indicator. The value of composite reliability varies between 0 and 1, with a better value indicating a higher level of reliability. In exploratory research, the value of composite reliability between 0.6–0.7 is acceptable, whereas the value of 0.7–0.9 is satisfactory [53,61].

The model testing in this article consists of two steps, namely testing the measurement model and testing the structural model. Testing the measurement model aims to make sure that the analysis instrument is reliable and valid. Structural model testing was conducted to examine the connection between dependent and independent variables. Testing is done through the Smart-PLS software. The relationship of the structural model in PLS-Sem, i.e., the disposition of dependent and independent variables, is shown in Figure 2.

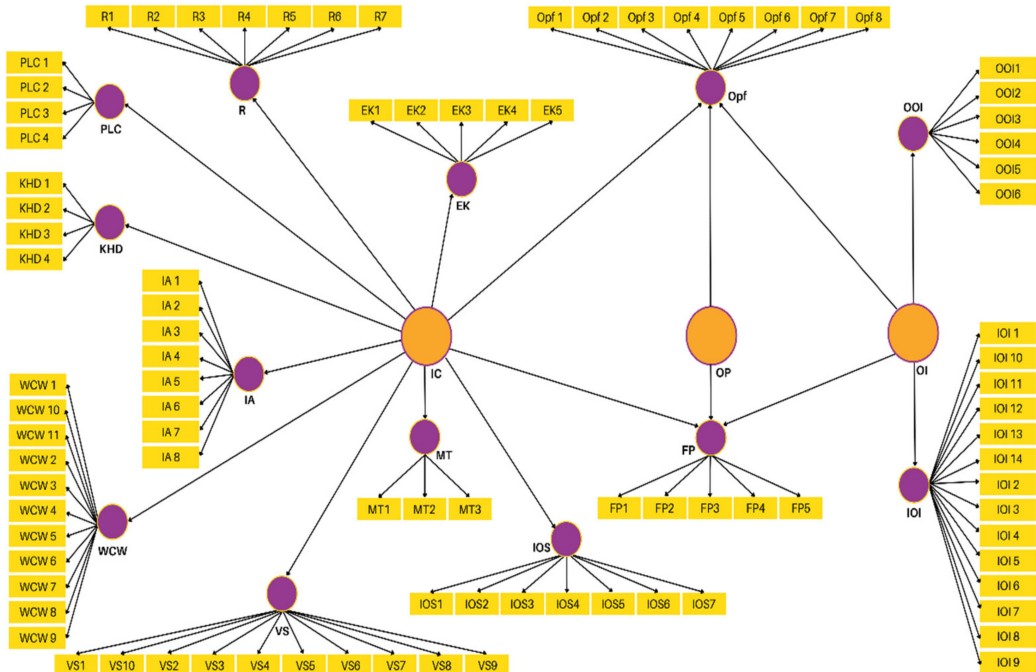

**Figure 2.** Relationship the structural model in PLS-SEM.

## 4. Result

This section reports on the results of the investigation through the PLS-SEM technique. To do this, the measurement model is first analyzed and the structural model later.

### 4.1. Measurement Model

Model testing can be evaluated through four criteria, i.e., composite reliability to measure the reliability of internal consistency, outer loading, and average variance extracted (AVE) to measure the convergent validity and cross-loading and Fornell-Larcker to evaluate the discriminant validity [62]. The indicator used to measure the reliability of internal consistency is alfa Cronbach, which is based on the indicator's inter-correlation, assuming that all indicators have the same outer loading on their construct. The value of composite reliability varies between 0 and 1, with a higher value indicating a higher degree of validity. In exploratory research, a composite reliability value of 0.6–0.7 can be accepted, while 0.7–0.9 is considered satisfactory [62]. The convergent validity examines to what extent a measurement positive correlates with the alternative measurement of the same construct. Indicators in the reflective construct are treated as different approaches to measuring the same construct, which is why indicators for a construct need to share a high variance proportion. The criteria to examine the convergent validity are the indicator's outer loading and average variance extracted (AVE) [62,63]. A high outer loading value signifies that

the involving indicators have a lot of similarities. All the outer loading values must be significant. Even though from the rule of thumb, the outer loading value is expected to be at least 0.7–0.8, the range between 0.4 and 0.7 can still be maintained. The latter range can still be considered without being eliminated from the measurement only if the composite reliability or AVE is beyond the threshold. Indicators with outer loading values below 0.4 can be eliminated from the measuring scale [62]. Discriminant validity examines to what extent a construct is different from other constructs by an empirical standard. The criteria to measure discriminant validity is cross loading value and Fornell-Larcker value [62]. Cross-loading value projects the indicator's outer loading to all constructs. The indicator's outer loading value of a related construct must be larger than the indicator's outer loading of other constructs [60,61]. Meanwhile Fornell–Larcker value observes the square comparison from AVE value against the correlation of latent variable. Square value of AVE must be larger than its highest correlation with other constructs [61].

The result from the model suggests that all outer loading values are more significant than 0.4; the composite reliability and AVE for all constructs are greater than 0.5; the Fornell-Larcker value for regenerations is smaller than the correlation between innovation capability and regeneration, and therefore the worth for open innovation is additionally smaller than the correlation between open innovation and outbound open innovation. For the Fornell-Larcker value, the correlation value that is smaller between construct and latent variable is often disregarded due to both variables representing the connection between the variable and its dimension. All cross-loading values every indicator on each variable is beyond the values for other variables. Supported the calculation results, it can be said that in terms of the perspective of convergence and discriminant, the model is reliable and valid. Table 1 presents the composite reliability value and Average Variance Extracted (AVE).

**Table 1.** Composite Reliability Value and Average Variances Extracted.

| Measuring Instrument | Alfa Cronbach | Composite Reliability | Average Variance Extracted (AVE) |
|---|---|---|---|
| Financial Performance (FP) | 0.757 | 0.757 | 0.892 |
| Operational Performance (OpF) | 0.896 | 0.897 | 0.928 |
| Inbound Open Innovation (IOI) | 0.821 | 0.841 | 0.869 |
| Outbound Open Innovation (OOI) | 0.633 | 0.721 | 0.505 |
| Participatory Leadership Culture (PLC) | 0.795 | 0.787 | 0.725 |
| Ideation and Organizing Structures (IOS) | 0.887 | 0.874 | 0.806 |
| Work Climate and Wellbeing (WCW) | 0.710 | 0.897 | 0.762 |
| Know-how Development (KHD) | 0.629 | 0.777 | 0.530 |
| Regeneration (R) | 0.781 | 0.798 | 0.867 |
| External Knowledge (EK) | 0.767 | 0.801 | 0.696 |
| Individual Activity (IA) | 0.733 | 0.849 | 0.652 |
| Vision and Strategy (VS) | 0.745 | 0.854 | 0.661 |
| Management of Technology (MT) | 0.795 | 0.807 | 0.712 |

*4.2. Structural Model*

Based on the relationship analysis presented in Table 2, it can be implied that there is a significant relationship between innovation capability (IC) towards financial performance (FP) and operational performance (OpF); the same goes to open innovation (OI) towards financial performance (FP) and operational performance (OpF). All hypotheses are accepted statistically despite differences in significance level. Hypothesis 1 (H1) is significant at alpha value 0.001; Hypothesis 2 (H2) is significant at alpha value 0.1; Hypothesis 3 (H3) and Hypothesis 4 (H4) are both significant at alpha value 0.05. All hypotheses are arranged in the model because they are based on the need for adjustment in the modeling of SMEs objects in Indonesia. To find out the close relationship between the variables in the hypothesis, so we use Pearson Coefficient Correlation. The result of the Pearson correlation for Hypothesis 1 (H1) is 0.889, which indicates the high relationship between innovation

capability and a firm's operational performance. In Hypothesis 2 (H2) relationship between Innovation capability and a firm's financial performance is 0.912. Hypothesis 3 (H3) testing shows that the relationship between open innovation and open performance is 0.966, while Hypothesis 4 (H4) states that the relationship between open innovation variables on the financial performance of 0.862. From these correlation value, all the result value is above 0.8, which means that there is a strong relationship between the variables in the hypothesis. So, these results strengthen the condition that H1, H2, H3, and H4 are eligible for the model.

**Table 2.** Significance of Structure Relationship.

| Hypothesis | | Path Coefficient | T-Statistic | *p* Value | Conclusion |
|---|---|---|---|---|---|
| H1 | IC → FP | 0.925 | 52.872 | 0.000 | Accept **** |
| H2 | IC → OpF | 0.620 | 10.290 | 0.004 | Accept ** |
| H3 | OI → FP | 0.823 | 31.330 | 0.001 | Accept *** |
| H4 | OI → OpF | 0.819 | 17.518 | 0.002 | Accept *** |

Note(s): **** significant at level 0.01; *** significant at level 0.05; ** significant at level 0.10.

The $R^2$ value, as shown in Table 3, illustrates how the variance of the endogen variable can be, for the most part, explained by its exogen variable. There are, however, two exogen variables, i.e., outbound open innovation and know-how development, that cannot explain their endogen variables. Outbound open innovation has an $R^2$ value of 0.244 while know-how development was at 0.311—both of which are considered moderate, which means they can partially explain their respective endogen variable. Meanwhile, the $R^2$ value of other exogen variables was greater than 0.5, meaning that they can explain their endogen variables well.

**Table 3.** $R^2$ Value for SMEs.

| | R Square | R Square Adjusted |
|---|---|---|
| Financial Performance (FP) | 0.944 | 0.944 |
| Operational Performance (OpF) | 0.717 | 0.707 |
| Inbound Open Innovation (IOI) | 0.998 | 0.898 |
| ***Outbound Open Innovation (OOI)*** | ***0.249*** | ***0.244*** |
| Participatory Leadership Culture (PLC) | 0.710 | 0.706 |
| Ideation and Organizing Structures (IOS) | 0.616 | 0.611 |
| Work Climate and Wellbeing (WCW) | 0.543 | 0.541 |
| ***Know-how Development (KHD)*** | ***0.311*** | ***0.307*** |
| Regeneration (R) | 0.532 | 0.528 |
| External Knowledge (EK) | 0.896 | 0.896 |
| Individual Activity (IA) | 0.927 | 0.926 |
| Vision and Strategy (VS) | 0.904 | 0.903 |
| Management of Technology (MT) | 0.527 | 0.522 |

Table 4 presents the significant result for each dimension for the dependent variable organizational performance. The dimension of financial performance significantly explains organizational performance, and the same goes for operational performance, which significantly explains organizational performance. Table 5 present the significant result for the variable innovation capability. Each dimension of innovation capability in Table 5 shows a significant relationship with the innovation capability variable.

**Table 4.** Relationship Significance for Organizational Performance with Its Dimension.

| Relationship | Correlation Value | T-Statistic | *p*-Value | Conclusion |
|---|---|---|---|---|
| *Organizational Performance →  Financial Performance* | 0.954 | 12.277 | 0.002 | Significant |
| *Organizational Performance →  Operational Performance* | 0.971 | 20.290 | 0.000 | Significant |

**Table 5.** Relationship Significance for Innovation Capability with Its Dimension.

| Relationship | Correlation Value | T-Statistic | *p*-Value | Conclusion |
|---|---|---|---|---|
| *Innovation Capability* → *Participatory Leadership Culture* | 0.964 | 5.465 | 0.004 | Significant |
| *Innovation Capability* → *Ideation and Organizing Structures* | 0.959 | 8.221 | 0.002 | Significant |
| *Innovation Capability* → *Work Climate and Wellbeing* | 0.771 | 17.895 | 0.000 | Significant |
| *Innovation Capability* → *Know-how Development* | 0.671 | 10.792 | 0.001 | Significant |
| *Innovation Capability* → *Regeneration* | 0.816 | 6.727 | 0.003 | Significant |
| *Innovation Capability* → *External Knowledge* | 0.946 | 53.115 | 0.000 | Significant |
| *Innovation Capability* → *Individual Activity* | 0.954 | 60.488 | 0.000 | Significant |
| *Innovation Capability* → *Vision and Strategy* | 0.949 | 56.269 | 0.000 | Significant |
| *Innovation Capability* → *Management of Technology* | 0.859 | 12.618 | 0.001 | Significant |

Table 6 present the significant result for the variable open innovation. This table describes that every dimension of the variable open innovation also features a significant relationship—in other words, those dimensions can be used to measure the variable. Figure 3 describes the structural relationship in the research model that depicts all hypotheses that have a significant relationship with their respective dependent and independent variable.

**Table 6.** Relationship Significance for Open Innovation with Its Dimension.

| Relationship | Correlation Value | T-Statistic | *p*-Value | Conclusion |
|---|---|---|---|---|
| *Open Innovation* → *Inbound open innovation* | 0.993 | 30.691 | 0.000 | Significant |
| *Open Innovation* → *Outbound open innovation* | 0.865 | 7.235 | 0.001 | Significant |

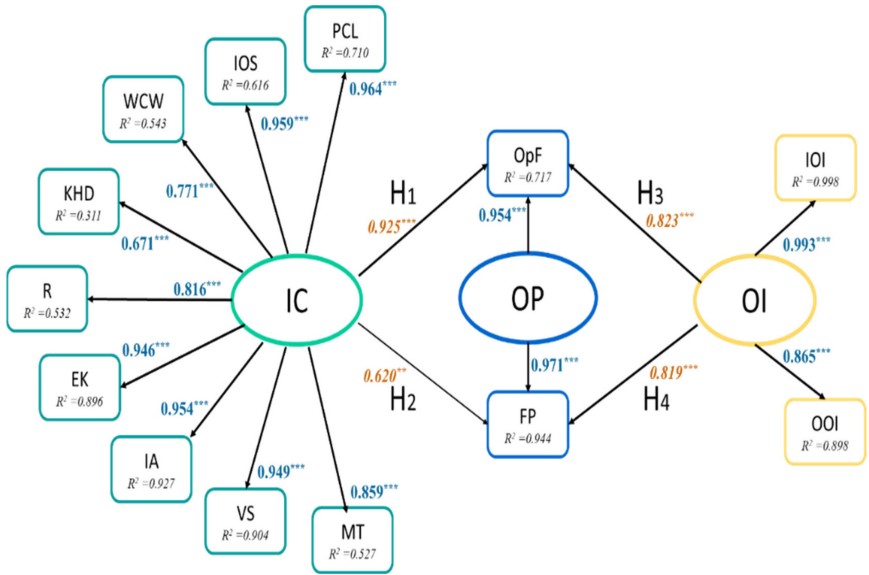

**Figure 3.** Relationship significance in the structural model. Note(s): *** *significant level α = 0.01;* ** *significant level α = 0.05.*

## 5. Discussion

This section discusses the main findings of the research study based on the theory of innovation capability, open innovation, and firm performance or organizational performance. In order to answer the objectives and questions of the investigation, the results derived from the verification of the hypotheses of the proposed theoretical model are described below.

In the context of the crisis due to the COVID-19 pandemic, innovation has been identified as one of the strong triggers for organizations to survive in both the service and manufacturing sectors [62,63]. The crisis that occurred during the COVID-19 pandemic was different from the previous crisis, so it would have had more of an impact on each country's economy [64,65]. The pandemic crisis is characterized by its unpredictable duration and its effects on various fields, especially the economic sector [66]. All business actors, including SMEs feel this impact. Small and medium enterprises (SMEs) play an important role in the economy, especially in developing countries [65]. The effect of the COVID-19 pandemic has had a significant impact on the SME economy. The worst impact is that most of these SMEs are forced to close their businesses [67,68].

The COVID-19 pandemic has become one of the driving forces for the business world to be motivated to improve their organizational performance, especially in SMEs, in developing countries [67]. SMEs are encouraged to be able to produce relevant innovations to overcome the negative consequences of the pandemic. COVID-19 has increased the mindset of SMEs leaders to maintain the performance of their SMEs, especially by involving their stakeholders due to the limited resources they have [69]. In this study, it can be shown that the performance of SMEs organizations is significantly affected by the ability of these SMEs to manage their innovation capability and open innovation processes, especially during a crisis due to the current pandemic.

The findings of this study have implications regarding the importance of implementing the proposed model in improving organizational performance in SMEs through open innovation activities and better innovation capability so that they can respond to the negative impacts caused by the COVID-19 pandemic. The results of this study show SMEs the importance of optimal organizational performance as a survival strategy and to strengthen them in facing future crises [70].

The results of the SEM analysis in this study indicate that innovation capability and open innovation are factors that significantly influence the performance of SMEs. This is, of course, one of the findings that are quite important in the survival of SMEs, especially in the face of the COVID-19 pandemic situation. Several dimensions of innovation capability that explain the most influential dimensions are Participatory Leadership Culture, Ideation and Organizing Structures, and Individual Activity. While in open innovation, the dimension that gives the most significant influence in inbound open innovation.

This study supports the hypothesis formulated in Hypothesis 1 (H1) and Hypothesis 2 (H2), which states that innovation capability has a significant influence on organizational performance, namely on financial performance and operational performance. This shows that innovation capability is a concern for SMEs in maintaining their business, especially in facing the impact of the COVID-19 pandemic. Strengthening the dimensions that support innovation capability needs to be improved in various aspects of activities in these SMEs.

In addition, through this research, it can also be empirically proven that open innovation has a significant influence on the performance of SMEs, both for financial performance and operational performance. Through hypothesis 3 (H3) it can be proven that open innovation has a significant effect on financial performance, while through Hypothesis 4 (H4) it is also proven that open innovation has a significant effect on operational performance. The dimension of open innovation that gives the most significant influence is inbound open innovation. This shows that activities related to inbound open innovation need to acquire considerable attention from the leadership of SMEs. The results of Hypothesis 3 and Hypothesis 4 indicate that SMEs need to increase their open innovation activities by exploring their innovation resources, both within the organization and outside the

organization. In this case, the role of stakeholders is quite important in providing support for the open innovation process in SMEs, especially in facing difficult times due to the COVID-19 pandemic and in being able to continue to survive in the uncertain economic situation due to the pandemic.

Empirical results in this study indicate that SMEs innovation capability and open innovation have a significant effect on the performance of these SMEs. This is in accordance with previous research, which states that innovation capability affects the performance of an organization [3,4], and open innovation has a significant impact on company performance [2,25]. There are several dominant dimensions in innovation capability, with a correlation value of more than 0.95, namely participatory leadership culture, ideation, organizing structures, and individual activities. While for open innovation, open inbound innovation is a more dominant dimension than outbound open innovation, with a value of 0.993. This research implies that SMEs need to make more aggressive efforts in carrying out activities related to dominant dimensions that significantly influence the achievement and improvement of innovation capability and open innovation. Organizational performance can be identified through one's financial condition and ability to produce goods and services—operational performance [2]. Therefore, according to the results of this study, SMEs also need to pay attention to the achievement of organizational performance through two main dimensions, namely operational and financial. Through the results of this study, financial performance has a more significant influence on organizational performance. This encourages SMEs to carry out various innovations to increase their overall business size and profitability.

## 6. Conclusions and Recommendation

In this study, a more comprehensive approach was conducted by simultaneously analyzing two main variables, namely innovation capability and open innovation, to achieve a more optimal organizational performance. The word optimal serves under the context of a more challenging business environment during the pandemic. Based on the results of this study, it can be empirically proven that innovation capability and open innovation affect organizational performance. In this article, the research model consists of three main concepts, namely innovation capability, open innovation, and organizational performance—where construction is studied in the context of SMEs, using purposive sampling technique, obtained a sample of 206 SMEs. The dimensions that give dominant influence on the innovation capability are participatory leadership culture, the most significant relationship with a correlation value of 0.964, followed by ideation and organizing structures with a correlation value 0.959, and individual activity with a correlation value of 0.954. For open innovation, inbound open innovation is greater than outbound open innovation, with a correlation value of 0.993. As for organizational performance, operational performance provides a more significant correlation with a correlation value of 0.971.

Overall, the results of this article have important implications and make a significant contribution, especially for the development of SMEs. The implications of this research are applied to batik SMEs, namely the ability to innovate and open innovation can facilitate greater organizational performance. These SMEs need to look for initiatives that can strengthen open innovation, where open innovation has a higher significance than outbound open innovation. To strengthen open inbound innovation, SMEs can collaborate with the authorities to support internal innovation efforts, for example with the government, other batik SMEs or other stakeholders. The interaction should go both ways, meaning the government should be more proactive in generating or scaling up efforts that enhance SME innovation efforts through knowledge sharing and other capacity building that helps enrich the flow of knowledge to these SMEs. In addition, batik SMEs can also have virtual discussions with suppliers, customers and other parties to discuss current challenges and initiatives to resolve these challenges—this session can also be used as part of branding to create brand awareness in the community.

The innovation capability is also proven to be a significant variable in realizing better organizational performance. A participatory leadership culture, as the strongest dimension for innovation capability, shows how important it is to develop leaders who can create a collaborative culture where employees are given the opportunity to share ideas and are treated as partners in directing the direction of the business. In addition, ideas and organizational structure are second, suggesting that SMEs need to build systems in which innovation-generating initiatives are activated and encouraged, for example, through incentive or reward systems.

The contribution of this article is to build an empirical model to assist SMEs in implementing the construct or variable of innovation capability and open innovation to achieve better organizational performance. To the authors' knowledge, no other studies have explored these variables overall in SMEs. The results of this research will then be disseminated as part of a training effort for batik SME owners and employees in the hope of improving the process of new ideas.

As any other research, there are some limitations which can be addressed in future research. Based on the results of the research that has been conducted, there are still several opportunities that can be developed for further research. There are several limitations in this study, the first being the respondents in SMEs. In this study, each SME was only represented by the owner or leader as a respondent. To increase representation, the research design can be improved by involving more key stakeholders in SMEs. This is related to the second limitation, namely the amount of data that is less than optimal. Even if it meets the test requirements, a larger number of respondents can increase the actual representation in the field, which results in more accurate generalizations. The limited amount of data is caused by the availability of time, environmental conditions, and health protocols that must be adhered to during the pandemic. The third limitation is that the type of respondents in this study is still targeting batik SMEs with one-time data analysis where cross sectional static data is taken for all variables. This means changes in the context and nuances of the organizational background cannot be captured. It is recommended that other types of SMEs be included in future research, as well as conducting longitudinal studies to ensure consistency of results across sectors. In further research, respondents can be distinguished based on their geographical location to identify directly the nuances that can affect readiness for open innovation. Further researchers can consider these suggestions to increase the validity of the results of this study.

**Author Contributions:** A.A.R. is the author of this manuscript and all of its contents and was responsible for designing the instrument and collecting and analyzing data in this investigation. Conceptualization, A.A.R.; methodology, A.F.R.; software, A.A.R. and A.F.R.; validation, A.A.R.; formal analysis, A.A.R. and A.F.R.; investigation, A.A.R.; resources, A.A.R.; data curation, A.A.R., A.F.R. and L.S; writing—original draft preparation, A.A.R.; writing—review and editing, L.S. and R.R.; visualization, L.S.; supervision, A.A.R.; project administration, A.F.R. and L.S.; funding acquisition, M.M.I. All authors have read and agreed to the published version of the manuscript.

**Funding:** This research received no external funding.

**Institutional Review Board Statement:** Not applicable.

**Informed Consent Statement:** Informed consent was obtained from all subjects involved in the study.

**Data Availability Statement:** The data and the questionnaire used in the study are available to other authors who require access to this material.

**Acknowledgments:** The author would like to thank the owners of batik SMEs who have helped and supported this research with valuable data, information and new insights, which can complement and enrich the studies in this research.

**Conflicts of Interest:** The authors declare no conflict of interest.

**Appendix A**

**Organizational Performance**

Financial Performance (FP) A measure of how performance is doing from the financial perspective

| | |
|---|---|
| FP1 | Organization considers the trend of net profit as a performance measurement |
| FP2 | Organization's net profit has increased consistently over the past year |
| FP3 | Organization considers the surge of revenue as a performance measurement |
| FP4 | The surge of organization's revenue has been consistent over the past year |
| FP5 | Organization compares the investment cost against the value it generates. |

Operational Performance (OpF)

A measure of how performance is doing from the operational perspective

| | |
|---|---|
| OpF1 | Customer feels satisfied with the product and service from the organization |
| OpF2 | Organization ensures employees' spirit remain high particularly during the pandemic |
| OpF3 | The productivity target set by the organization has always been achieved over the past year |
| OpF4 | The quality of product made by the employee is up to the pre-determined standard |
| OpF5 | The product or service is delivered to the customer in a timely manner |
| OpF6 | The book-keeping process of inventory is done properly |
| OpF7 | The product made by organization dominates the market |
| OpF8 | Total sales increased over the past year |

**Innovation Capability**

Participatory Leadership Culture (PLC)

The culture within the organization that is inclusive in its member's participation

| | |
|---|---|
| PLC1 | Manager allows subordinates to participate in the product development process |
| PLC2 | The inclusion of employee's opinion could help organization shape the innovation direction |
| PLC3 | The leadership style in the organization is not merely instructing but open for collaboration |
| PLC4 | The leader shows support to critical ideas that aim to reform the organization |

Work Climate and Wellbeing (WCW)

The culture within the organization that is inclusive in its member's participation

| | |
|---|---|
| WCW1 | Integrity in working is important for the organization |
| WCW2 | Competency in working is important for the organization |
| WCW3 | Reliability in working is important for the organization |
| WCW4 | Loyalty in working is important for the organization |
| WCW5 | Openness to others in working is important for the organization |
| WCW6 | Organization sees every member as equal |
| WCW7 | Everyone understands their role within the organization |
| WCW8 | Organization demonstrates effort to improve employee's creativity in Working |
| WCW9 | Organization demonstrates effort to improve employee's independence in working |
| WCW10 | The leader models behaviors that empower employees |
| WCW11 | Every employee is treated fairly |

Ideation and Organizing Structure (IOS)

The process of gathering ideas, and the organizational structure

| | |
|---|---|
| IOS1 | Work activities within organization are conducted in an organized manner |
| IOS2 | There is a procedure that facilitates innovation within the organization |
| IOS3 | Organization has within it a formal position responsible for innovation process |
| IOS4 | Organization has tools to facilitate idea generation process |
| IOS5 | The organization structure is informal |
| IOS6 | The decision can be made by someone other than the owner |
| IOS7 | Organization implements a reward system for creative and innovative initiatives |

Know-How Development (KHD)

The process of developing practical knowledge in working

| KHD1 | Organization gives training to improve employee's skills to understand customer's situation |
| KHD2 | Organization gives training to improve employee's skills to understand competitor's situation |
| KHD3 | Organization gives training to improve employee's skills to understand the emergence of new technologies |
| KHD4 | Organization gives supplementary training to maximize employee's potential |

### Regeneration (Reg)
The process of creating newness in the organization

| Reg1 | Organization conducts an evaluation to assess the effectiveness of business initiatives |
| Reg2 | Organization implements changes based on prior evaluation |
| Reg3 | Manager treats mistakes as a learning opportunity for employees |
| Reg4 | Organization perceives innovation as essential |
| Reg5 | Organization is willing to take business risk |
| Reg6 | Organization is willing to exchange opinion |
| Reg7 | There is trust and respect with each other in the organization |

### External Knowledge (EK)
The process of acquiring knowledge from external parties

| EK1 | Organization collaborates with external parties |
| EK2 | Organization has a good relationship with suppliers |
| EK3 | Organization has a good relationship with customers |
| EK4 | Organization has a good relationship with industrial associations |
| EK5 | Organization has a good relationship with competitors |

### Individual Activity (IA)
Activities of each member of the organization that is potential to create innovation

| IA1 | Member of organization has new perspective on issues at organization |
| IA2 | Member of organization is willing to take risk |
| IA3 | Member of organization embraces uncertainty |
| IA4 | Member of organization has intrinsic motivation to work |
| IA5 | Organization changes individual tasks based on changes in business climate |
| IA6 | Organization provides time for member to nurture creativity |
| IA7 | Organization provides budget for member to nurture creativity |
| IA8 | Organization provides tools and setups for member to nurture creativity |

### Vision and Strategy (VS)
The direction and strategies taken by organization in order to compete in the market

| VS1 | The leader socializes the organization's vision and mission to all members |
| VS2 | Each member knows the organization's vision and mission |
| VS3 | Organization decides business and functions that are performed by them |
| VS4 | Organization decides business and functions that are not performed by them |
| VS5 | Organization decides the scope of their target market |
| VS6 | Organization can identify their unique value and differentiating factors compared to competitors |
| VS7 | There is a new creation of product and service to stimulate demand |
| VS8 | Organization is oriented towards the future |
| VS9 | There are initiatives to save cost |
| VS10 | There are initiatives to increase quality within organization |

### Management of Technology (MT)
The role of technology within the organization

| MT1 | Organization includes technology into their generic strategy |
| MT2 | Organization can anticipate the technological needs in the future based on the development of product and market trend |
| MT3 | Organization conducts assessment on technological needs |

**Open Innovation**
Inbound Open Innovation (Inb)

Innovation in organizations is distributed by exploring sources of innovation corresponding to knowledge and new technologies from external sources such as customers, suppliers, competitors, government, consultants, universities, or research organizations

| | |
|---|---|
| Inb1 | External parties who are directly involved in innovation activities within the organization |
| Inb2 | The government plays a job in helping innovation activities within the organization |
| Inb3 | Consumers contribute in innovation activities within the organization |
| Inb4 | Competitors contribute in innovation activities within the organization |
| Inb5 | Research institutions provide assistance in innovation activities within the organization |
| Inb6 | Universities or educational institutions contribute to innovation activities within the organization |
| Inb7 | The supplier contributes to the organization's internal innovation activities |
| Inb8 | There are consultants who provide assistance in innovation activities within the organization |
| Inb9 | Innovative activities meted out by the organization depend upon the help of external parties |
| Inb10 | I use the latest equipment to improve innovation within the organization |
| Inb11 | I use the newest materials (fabrics, dyes, candles, etc.) to enhance innovation within the organization |
| Inb12 | I bought patents for innovation activities within the organization |
| Inb13 | I bought copyright to be used for innovation activities within the organization |
| Inb14 | I bought a license to be used for innovation activities within the organization |

Outbound Open Innovation (Outb)

Innovation within the organization is meted out by developing internal organizational innovation capabilities to have an impact that can be given to external organizations through licenses, patents, or sure contracts to obtain financial and non-financial benefits.

| | |
|---|---|
| Outb1 | I try to get other benefits from the internal innovations that have been made |
| Outb2 | I offer new methods used by internal organizations in other organizations |
| Outb3 | The organization sells batik product licenses to other organizations |
| Outb4 | I sell batik product patents to other organizations |
| Outb5 | I sell copyrighted batik motifs to other organizations |
| Outb6 | I collaborated by selling the latest technology (for example, new batik tools, new batik ways, new waste process methods) to manufacture batik in different organizations. |

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
