# Peer review of "Innovation Capability and Open Innovation for Small and Medium Enterprises (SMEs) Performance: Response in Dealing with the COVID-19 Pandemic"

_sustainability, doi:10.3390/su14105874_

Round 1

Reviewer 1 Report

The issue analyzed is of high importance.

  1. The structure is interesting but lets a few aspects open. For example, considering "innovation capability" and "open innovation" as two independent variables is quite problematic.

2.What really is innovation capability?  Please give detailed  explaination of this. And is not open innovation a strong element of innovation capability?

3. Are those two variables actually independent, or are they themselves a function of organizational and financial strength?

4. The paper needs a very heavy language and style editing. 

Author Response

Response to Reviewer 1 Comments

Point 1: The structure is interesting but lets a few aspects open. For example, considering "innovation capability" and "open innovation" as two independent variables is quite problematic.

Response 1:

The explanation about "innovation capability" and "open innovation" in this study is revealed in line 162 -181.

Based on previous research from Lawson and Samson [4] and Minna [3] explained that innovation capability has an influence on the performance of an organization, while according to Popa et.al and Rumanti et.al it is explained that empirically open innovation has an influence on organizational performance, both in the form of operational performance or financial performance [2][23]. Therefore, this study was con-ducted to show how the influence of each construct, namely innovation capability and open innovation independently, on the performance of the organization/company. Rajapathirana et.al (2017) research also states that innovation capability is an important factor in organizational or company performance, especially financial performance [33]. In addition, Yusr (2016) also stated that innovation capability has a significant influence on innovation performance in organizations or companies [34]. Carjaval (2021) conveyed that the relationship between the construct of innovation capability to open innovation has a weak effect, which is less than 0.5 [35], as well as the relationship between the construct of open innovation to performance with a correlation of 0.16 [35]. Hoang et.al (2019) also states that there is a strong relationship or influence between innovation capability and organizational performance [36]. Therefore, in this study, a research model was constructed in which innovation capability and open innovation were directly related to organizational performance. Two constructs are independent variables on the basis that each construct has an influence on the performance of the organization or company, as has been conveyed through the model elaboration scheme in Figure 1.

In this study, it can be proven that innovation capability has a strong relationship with financial performance and operational performance, with a correlation value of 0.925 and 0.620. In addition, the open innovation construct also has a strong relationship with financial performance which is 0.823 and 0.819 with operational performance.

Point 2: What really is innovation capability?  Please give detailed explanation of this. And is not open innovation a strong element of innovation capability?

Response 2:

The detail explanation about innovation capability in this study is revealed in line 207 -235.

Innovation capability is the ability to shape and manage a wide range of capabili-ties. In an organization or company, innovation capability integrates the organiza-tion's key capabilities and resources to stimulate innovation successfully in an effort to achieve optimal organizational performance [40]. Innovation capability also reflects the ability to continuously transform knowledge and idea into a new product, process, and system for the benefit of the organization as well as its stakeholders. Innovation capability can be seen as a dynamic capability that shows an organization’s ability to integrate, build, and reconfigure both internal and external competence to deal with an environment that changes rapidly [41]. Innovation capability also acts as a funnel that can seek, locate and develop potential innovations to be streamed to the core process in the organization [4]. A high degree of innovation capability enables the organization to bring efficiency together with creativity [4] and helps businesses to have the ability to create and develop excellent new products, which is a critical aspect in order to sur-vive and prosper for a long term, it is critical for the business to have the ability to cre-ate and develop excellent new products [42]. That is why innovation capability is re-garded as a valuable asset for firms in terms of providing and sustaining competitive advantage and implementing corporate strategy, and it also helps the organization to form, manage, and integrate multiple capabilities or stimulus to innovation success-fully [39]. Innovation capability is considered one of the activities or policies of the or-ganization in improving its performance, including in SMEs. Several studies had pro-posed direct relationship of innovation capability to organizational performance [3] [34] [43] [44]. It indicates that innovation capabilities determine of firm performance [45]. Thus, the development and analysis of indicators in innovation capabilities is important in facilitating performance evaluation in SMEs [46][47]. It is critical because innovation is believed to be the most important driver of organizational performance and serves as a key role to the survival and growth of an organization [3]. It is believed that innovative firms would have better firm performance compared to those that do not innovate. Innovative organizations show a higher level of economic growth and productivity than non-innovative ones [33]. Moreover, the study provided a positive relationship between organizational innovation capability and performance.

Point 3: Are those two variables actually independent, or are they themselves a function of organizational and financial strength?

Response 3:

In this study, innovation capability and open innovation is proposed as determinant of organizational performance i.e. financial performance and operational performance. Both of the variables are proposed as independent variable, so the role of each variables to each type of performance can be studied, especially from Indonesian SMEs point of view. We expect that by treating those two variables as independent variable we can gain the insight regarding how each variable drive organizational performance. The result of this study shows that innovation capability has stronger effect towards operational performance, while its effect towards financial is considered lower. Moreover, the effect of open innovation practice towards operational and financial performance tend to have values that are not much different that implies that the role of open innovation to financial performance is as strong as the its role to operational performance.

Point 4: The paper needs a very heavy language and style editing.

Response 4: Thank you for your comment and advise. We have made a revision considering the comment by improve language in this manuscript.

Author Response

Response to Reviewer 2 Comments

Point 1: The topics related to organizational innovation capacity and open innovation practices are of interest from the point of view of functioning enterprises in a dynamically changing economic reality. Additionally, the Covid-19 Pandemic and Russia's current actions in Ukraine require swift and decisive organizational action to help businesses survive this difficult period. Therefore, I believe that the topics addressed in the manuscript are essential and may interest potential readers. The structure of the manuscript is correct. However, I have one comment for the authors that should be considered before publishing the manuscript. In the theoretical part of the manuscript, the authors discuss the ability of organizations to innovate and emphasize that the idea of innovating improves organizational performance in various ways. However, the organization itself does not create or adopt innovations; people do. The people who are members of the organization determine its innovation. Therefore, what is missing here is a thread about human factors that affect the adaptation of innovation. I propose to complete the theoretical part by using the current literature dealing with the issues of innovation adaptation:

https://doi.org/10.3390/su14010140; https://doi.org/10.1007/s10488-013-0486-4

https://doi.org/10.3390/su12208630; https://doi.org/10.1016/j.iedeen.2021.100169

Response 1: The explanation about human factors that affect the adaptation of innovation in this study is revealed in line 129-152.

Individual characters in an organization or company will have an impact on the in-novation process [26], including the process and implementation of open innovation. Individual attitudes and motivations in innovating will affect the results of open in-novation carried out by the company [26].  The process of adopting innovation by individuals or employees in an organization or company contributes to the organi-zational culture. Open innovation based on organizational culture will help organiza-tions to seize new opportunities, this is because human or individual factors have a role in determining the success of the innovation process so that the human factor cannot be ignored to get the optimal innovation process for the company [27].

The challenge for SMEs to optimize their performance has become even more dif-ficult with the rise of the global pandemic of Covid-19 [28]. This happens to all organ-izations across the economic scale. In the innovation process in an organization or company, the adaptability of individuals or employees in the company is important, especially from the perspective of implementing new technologies and to be able to maintaining the ability to compete in today's dynamic market. During the COVID-19 pandemic, the implementation of open innovation is a challenge for organizations or companies. The negative impact of the pandemic is not only on the economic system but also on various social aspects of society that may not have been imagined before will become a global crisis [1].  For SMEs, the role of individuals is very im-portant in determining the readiness and success of these SMEs to carry out open in-novation, especially the ability to respond efficiently to international market dyna-mism. Human capital is essential for the development and commercialization of in-novations and plays a key role in the survival and success of the innovation process, especially in the face of today's seemingly dynamic market environment [29].

Reviewer 3 Report

The authors brought an interesting and actual topic. The empirical connection between open innovation and firm performance is vastly explored in the literature, still, the interconnectedness to innovation capability is overlooked. 

As a first and general comment, I would like to underline the importance of performing deep proofreading and rephrasing the document as there are typos, language flaws, and scant scientific soundness. 

Also, the connection to the pandemic crisis is, in my perspective, not grounded in the literature, as the establishment of linkages with external sources of knowledge is precedent to the crisis and has not changed significantly as the exogenous shock was in the demand. 

On the other hand, open innovation is approached from an old-fashioned perspective, considering the players in the value chain. However, the conceptual framework was revisited several times, and, at present, the reasonable paradigm is "open innovation 4.0" involving the user community and the environment. In this vein please consider: Costa, J.; Matias, J.C.O. Open Innovation 4.0 as an Enhancer of Sustainable Innovation Ecosystems. Sustainability 202012, 8112. https://doi.org/10.3390/su12198112. 

The user community is of particular interest as the new policy frameworks must enroll consumers to accelerate the introduction of responsible improvements in goods and services. This player adds consciousness to the innovative process.

In regards to the empirical part there are some flaws deserving consideration: 1) the number of responses does not grant any kind of representativeness, therefore the interpretation of the results needs to be carefully addressed. Also, there is a need to explain the sampling type as well as the selection criteria (convenience samples not seldom lack robustness). Hypotheses are incorrectly formulated - "significantly" does not specify the direction of the effect. Please consider changing.

There is no evidence to support H3 and H4 - the operational and financial performance have a multiplicity of drivers, needing other techniques to be proved as formulated e.g. propensity score matching. 

There is a need to enlighten the reader about the representativeness or the singularities of these firms to grant the validity of the results to the scientific community. 

It is very difficult to draw conclusions given the potential weakness of the respondent sample. 

Perhaps the authors should consider grounding the sector in specific literature, update the conceptual frameworks and explore the answers received in a contextualized vein. 

Best of luck with your research!

Author Response

Response to Reviewer 3 Comments

Point 1  : As a first and general comment, I would like to underline the importance of performing deep proofreading and rephrasing the document as there are typos, language flaws, and scant scientific soundness.

Response 1:

Thank you for your comment and advise. We have made a revision considering the comment by improve language in this manuscript.

Point 2: The connection to the pandemic crisis is, in my perspective, not grounded in the literature, as the establishment of linkages with external sources of knowledge is precedent to the crisis and has not changed significantly as the exogenous shock was in the demand.

Response 2:

The explanation about grounded literature in this study is revealed in:

Line 37-43

In the current era of the COVID-19 pandemic that has hit the whole world glob-ally, the implementation of open innovation for organizations is a challenge, especially because the negative impact of the pandemic is not only on the economic system but also on the order of people's lives in general. This has a negative impact that has never happened before a global crisis such as the current pandemic [1]. Referring to the cur-rent situation, open innovation is seen as a key aspect of economic recovery during and after the ongoing pandemic [1]. 

Line 83-98

This study was conducted to accommodate the current global pandemic, because during the current pandemic, SME owners or leaders are required to be able to main-tain their existence or sustainability by taking various ways to maintain their business performance, especially for SMEs in developing countries [17]. SMEs are an important factor in driving the economy and are one of the indicators that show the economic development of a country [18][19]. SMEs make a significant contribution to a country's GDP, both for developed and developing countries [17-19]. Therefore, the performance of SMEs is an important thing to study related to the pandemic conditions. In Indone-sia, SMEs are one type of business that is quite "tough" in facing the economic impact of the current pandemic. In addition, SMEs are an economic sector that absorbs much labor, so the existence of SMEs can reduce unemployment, especially because of the economic impact of the economic crisis due to the pandemic [17]. Seeing the im-portance of the existence of SMEs both in developing countries and in developed countries, the performance of SMEs needs to receive more in-depth attention and study, especially in the current pandemic conditions. Where SMEs are the backbone of a country's economy [20], and their performance is one of the factors that measure the success of economic activities in a country [18], [20].

Point 3: Open innovation is approached from an old-fashioned perspective, considering the players in the value chain. However, the conceptual framework was revisited several times, and, at present, the reasonable paradigm is "open innovation 4.0" involving the user community and the environment. In this vein please consider: Costa, J.; Matias, J.C.O. Open Innovation 4.0 as an Enhancer of Sustainable Innovation Ecosystems. Sustainability 2020, 12, 8112. https://doi.org/10.3390/su12198112. The user community is of particular interest as the new policy frameworks must enroll consumers to accelerate the introduction of responsible improvements in goods and services. This player adds consciousness to the innovative process.

Response 3:

The explanation about open innovation and the user community by considering: Costa, J.; Matias, J.C.O. Open Innovation 4.0 as an Enhancer of Sustainable Innovation Ecosystems. Sustainability 2020, 12, 8112. https://doi.org/10.3390/su12198112  is revealed in line 101-116.

Open innovation currently has a paradigm that adopts several influences from the ex-istence of digital technology and the global transformation of information and knowledge. This open innovation was later known as open innovation 4.0 [21]. Open innovation 4.0 provides space for all stakeholders involved to contribute more opti-mally to the company or organization. The transformation of information and knowledge, as well as its digitization is very important in the implementation of open innovation 4.0, Therefore, global collaboration networks and activities significantly impact open innovation 4.0. The open innovation 4.0 framework involves the customer community as an important factor in the improvements made by a company or or-ganization through open innovation activities. The involvement of the consumer community will accelerate the process of improvement and open innovation because it will increase the awareness of a company or organization of the importance of the in-novation process. The company will develop a new capability to engage the consumer community through digital media to collect and analyze data that is part of a dynamic ecosystem in a value chain [21]. This has an impact on the open innovation process for both large, medium, and small-scale companies, including SMEs. 

Point 4: In regards to the empirical part there are some flaws deserving consideration: 1) the number of responses does not grant any kind of representativeness, therefore the interpretation of the results needs to be carefully addressed. Also, there is a need to explain the sampling type as well as the selection criteria (convenience samples not seldom lack robustness). Hypotheses are incorrectly formulated - "significantly" does not specify the direction of the effect. Please consider changing.

Response 4:

The explanation about data sampling, data collection and the hypotheses are conveyed in:

Line 298-319

Data collection was carried out in two stages. The first stage is a preliminary study (pilot test) to test the validity and reliability of the measuring instrument in this study. In this preliminary stage, questionnaires were distributed to 30 respondents/SMEs who were randomly selected from all data on SMEs batik Lasem, Rembang Region, Central Java Province, and Madura, East Java Province. Based on the responses to the ques-tionnaire obtained in the preliminary study, several improvements were made to the questionnaire. Based on the initiation of the questionnaire that has been collected, sev-eral statements or questions must be modified in the phrase structure. This improve-ment is intended so that respondents can understand the intent of the questions asked. The improvement in question is to change the explanation in the narrative in the statement on the questionnaire so that respondents can better understand the question and provide accurate answers. Questionnaire responses obtained from the initial analysis were not included in the final sample of the study. The second stage is the col-lection of SMEs data which is carried out between June and August 2021. The popula-tion in this study is all SMEs in Lasem, Rembang Region, Central Java Province, and Madura, East Java Province. Due to the wide population coverage, the researchers chose to determine the sample based on SMIs affected by the Pandemic, characteristics such as: (i) Decreased sales results, (ii) Decreased number of employees, (iii) Average asset value of less than 1 billion, and (iv) age leadership over 40 years. So, in this study, the data collection technique used is purposive sampling, where the aim is to obtain a sample that can describe the population and also save costs and time. The number of samples taken from the population is five to ten times the number of variables used in the analysis design and a minimum of 200 samples [55].

Line 238-241

H1: There is a relationship between Innovation capability and a firm’s operational performance

H2:        There is a relationship between Innovation capability and a firm’s financial performance 

Line 276-278

H3: There is a relationship between open innovation and a firm’s operational performance

H4: There is a relationship between open innovation and a firm’s financial performance

Point 5: There is no evidence to support H3 and H4 - the operational and financial performance have a multiplicity of drivers, needing other techniques to be proved as formulated e.g. propensity score matching.

Response 5:

Evidence to support the hypotheses are explained in Line 350-361.

All hypothesis are arranged in the model because they are based on the needs of the adjustment in the modeling of SMEs objects in Indonesia. To find out the close relationship between the variables in the hypothesis, so we use Pearson Coefficient Correlation. The result of the Pearson correlation for Hypothesis 1 (H1) is 0.889, which indicates the high relationship between innovation capability and a firm's operational performance. In Hypothesis 2 (H2) relationship between Innova-tion capability and firm's financial performance is 0.912. Hypothesis 3 (H3) testing shows that the relationship between open innovation and open performance is 0.966, while Hypothesis 4 (H4) states that the relationship between open innovation varia-bles on the financial performance of 0.862. From these correlation value, all the result value is above 0.8, which means that there is a strong relationship between the varia-bles in the hypothesis. So, these results strengthen the condition that H1, H2, H3, and H4 are eligible for the model.  

Point 6: There is a need to enlighten the reader about the representativeness or the singularities of these firms to grant the validity of the results to the scientific community.

Response 6:

The explanation about the representativeness or the singularities of these firms to grant the validity of the results to the scientific community conveyed in:

Line 60-64

Successful innovation can bring uniqueness to the organization unavailable in other organizations, which becomes a source of advantage [8][9]. Innovation can also be seen as a process to increase the organization’s must-have capabilities regardless of the organization’s scale [8]. Innovation pertains to a continuous improvement within the organization to optimize its performance [10]

Line 71-73

According to the Statistic Center Indonesia 2020, there was a decrease in the number of SMEs in data collection locations by 7.06%, and by the end of 2020, there were 11.25% of SMEs not operating [13]. 

Line 402-405

Based on the data in Figure 3, most SME owners are aged more than 40 years but below 60 years old which represents 67%, i.e., 35% are between 40-49 years old and 32% are between 50-59. The minorities are those aged below 40 years old – where 20-29 years old only cover for 2 % and those between 30-39 are at 13%, and those above 60 years old who cover for 18% of respondents.

In addition, data collected in this study obtained from several regions in Central Java Province and East Java Province, where both province are considered as one of province that is well known for Batik product. Moreover, in determining sample, we choose Batik SME that is affected by Covid-19 Pandemic and has characteristic such as several criteria i.e. (i) Decreased sales results, (ii) Decreased number of employees, (iii) Average asset value of less than 1 billion, and (iv) age leadership over 40 years (see: line 289-319). Thus, the result obtained seems able to represent the condition the condition experienced by Batik SME, especially in Indonesia.

Point 7: It is very difficult to draw conclusions given the potential weakness of the respondent sample.

Response 7:

The explanation about the respondent sample is conveyed in:

Line 504-508

In this article, the research model consists of three main concepts, i.e., innovation capability, open innovation, and organizational performance – where constructs are scrutinized under the context of SME, by using a purposive sampling technique, a sample of 206 SMEs was obtained, from the results of this sample collection, the Open Innovation model testing can be carried out and the results are obtained. 

Line 298-319

Data collection was carried out in two stages. The first stage is a preliminary study (pilot test) to test the validity and reliability of the measuring instrument in this study. In this preliminary stage, questionnaires were distributed to 30 respondents/SMEs who were randomly selected from all data on SMEs batik Lasem, Rembang Region, Central Java Province, and Madura, East Java Province. Based on the responses to the ques-tionnaire obtained in the preliminary study, several improvements were made to the questionnaire. Based on the initiation of the questionnaire that has been collected, sev-eral statements or questions must be modified in the phrase structure. This improve-ment is intended so that respondents can understand the intent of the questions asked. The improvement in question is to change the explanation in the narrative in the statement on the questionnaire so that respondents can better understand the question and provide accurate answers. Questionnaire responses obtained from the initial analysis were not included in the final sample of the study. The second stage is the col-lection of SMEs data which is carried out between June and August 2021. The popula-tion in this study is all SMEs in Lasem, Rembang Region, Central Java Province, and Madura, East Java Province. Due to the wide population coverage, the researchers chose to determine the sample based on SMIs affected by the Pandemic, characteristics such as: (i) Decreased sales results, (ii) Decreased number of employees, (iii) Average asset value of less than 1 billion, and (iv) age leadership over 40 years. So, in this study, the data collection technique used is purposive sampling, where the aim is to obtain a sample that can describe the population and also save costs and time. The number of samples taken from the population is five to ten times the number of variables used in the analysis design and a minimum of 200 samples [55]. 

Point 8: Perhaps the authors should consider grounding the sector in specific literature, update the conceptual frameworks and explore the answers received in a contextualized vein.

Response 8: The explanation about grounded literature in this study is revealed in Line 83-98.

This study was conducted to accommodate the current global pandemic, because during the current pandemic, SME owners or leaders are required to be able to main-tain their existence or sustainability by taking various ways to maintain their business performance, especially for SMEs in developing countries [17]. SMEs are an important factor in driving the economy and are one of the indicators that show the economic development of a country [18][19]. SMEs make a significant contribution to a country's GDP, both for developed and developing countries [17-19]. Therefore, the performance of SMEs is an important thing to study related to the pandemic conditions. In Indone-sia, SMEs are one type of business that is quite "tough" in facing the economic impact of the current pandemic. In addition, SMEs are an economic sector that absorbs much labor, so the existence of SMEs can reduce unemployment, especially because of the economic impact of the economic crisis due to the pandemic [17]. Seeing the im-portance of the existence of SMEs both in developing countries and in developed countries, the performance of SMEs needs to receive more in-depth attention and study, especially in the current pandemic conditions. Where SMEs are the backbone of a country's economy [20], and their performance is one of the factors that measure the success of economic activities in a country [18], [20].

Round 2

Reviewer 1 Report

My comments for the previous version were not appropriately considered. As a whole, I think that the article does not yet reach the quality level sufficient for publication. 

Reviewer 2 Report

The authors made the necessary corrections.

Reviewer 3 Report

I would like to thank the authors for their effort in providing an improved version of the document. 

The first hindering factor I identify in the document has to do with the writing style and the scientific soundness - there is a need to improve the writing as there are unexpected elements in the phrases; also, some sentences have unnecessary information for a journal article: explaining in the abstract the use of a Likert scale in the questionnaire. 

At first, the title is too long, please reformulate it. In my view firm performance is not a strategy, it is an outcome of 

Also, the portrait of SMEs has some arguments which need further referencing (around line 120), it is true that SMEs may need more external information. Still, the literature agrees that they are more prone to have trouble being open. 

Around line 162 up to 182 the referencing does not follow the journal style. 

Around line 207 there is a definition I would like the authors to reconsider or rephrase: "Innovation capability is the ability to shape and manage a wide range of capabilities"  - there is no reference and, to me, it is not correct.

Then, around line 225 there is a mention of "organizational policies, please consider rephrasing. 

The conceptual model needs to be further justified as there is no solid link between the elements. There is a reference to several authors, but, the elements need to be intertwined. 

Please re-read and correct the extract between lines 308 and 319. It is not clear. 

The methodological section needs to be re-organized and the flow of the reading needs to improve. At first, the reader needs to understand why each step was given and the connection between them. The respondent sample should perhaps appear first to introduce the reader to what to expect. 

The respondents do not matter here as the study is focusing on the enterprise - please remove all information connected to this dimension.

The results need to be contextualized and further developed. 

The connections identified need to be tied to the previous literature. 

There is a need to include a solid conclusion section; e.g. contributions, improvements from previous studies, recommendations, limitations...... as the paper seems unfinished.  

Hypotheses are incorrectly formulated - establishing that there is a relationship is too vague, the direction of the effect needs to be identified. 
